# Integrated 3D printing of flexible electroluminescent devices and soft robots

Pei Zhang[1], Iek Man Lei[1], Guangda Chen[1], Jingsen Lin[1], Xingmei Chen[1], Jiajun Zhang[1], Chengcheng Cai[1], Xiangyu Liang[1] & Ji Liu [1,2,3] ✉

Flexible and stretchable light emitting devices are driving innovation in myriad applications, such as wearable and functional electronics, displays and soft robotics. However, the development of flexible electroluminescent devices via conventional techniques remains laborious and cost-prohibitive. Here, we report a facile and easily-accessible route for fabricating a class of flexible electroluminescent devices and soft robotics via direct ink writing-based 3D printing. 3D printable ion conducting, electroluminescent and insulating dielectric inks were developed, enabling facile and on-demand creation of flexible and stretchable electroluminescent devices with good fidelity. Robust interfacial adhesion with the multilayer electroluminescent devices endowed the 3D printed devices with attractive electroluminescent performance. Integrated our 3D printed electroluminescent devices with a soft quadrupedal robot and sensing units, an artificial camouflage that can instantly self-adapt to the environment by displaying matching color was fabricated, laying an efficient framework for the next generation soft camouflages.

The advent of flexible and stretchable electroluminescent (EL) electronics has enabled technological advances in myriad applications, such as information encryption[1–3], smart electronic skins[4–8], soft robotics[9–11], and optical communication[12,13]. Among these electroluminescent devices, alternating current EL (ACEL) devices are arguably one of the most suitable candidates for developing stretchable devices[14]. They feature not only simple architectures, promising ductility and robustness for harsh-environment applications, but also relatively facile manufacturing processes compared to the vapor deposition for organic-light emitting diode fabrication[5,7,10,11,15,16]. At present, flexible ACEL devices are commonly fabricated via multilayer lamination (i.e., screen printing), where an EL phosphor (such as ZnS:Cu dots) layer is sandwiched between two stretchable electrodes. However, the series of steps and expensive utilities (i.e., masks and delicate tools) required in this technique may limit its applications in rapid prototyping and customization[17]. With the increasing demand for innovation in flexible EL devices, a facile, easily-accessible, and customizable fabrication strategy is therefore urgently needed.

Multi-material 3D printing, on the other hand, is an emerging high-throughput and programmable manufacturing technique that enables the creation of multi-component 2D and 3D intricate objects from a wide range of functional viscoelastic materials, offering a viable strategy for achieving this goal[18–20]. However, despite recent progress in 3D printed electronics, such as displays, wearable electronics, solid-state lightings, and biomedical electronics[21,22], the fabrication of sophisticated EL devices through multi-material 3D printing remains largely unexplored[23–25]. Here, we report a streamlined approach for fabricating flexible EL devices through multi-material 3D printing (Fig. 1a). The device consists of a highly conducting elastomer as the electrodes, a dielectric elastomer as the insulating layer, and a ZnS phosphors-loaded elastomer as the electroluminescent layer. To achieve a printable system, the formulations of the inks were designed to exhibit favorable rheological properties for extrusion printing

[1]Department of Mechanical and Energy Engineering, Southern University of Science and Technology, Shenzhen 518055, China. [2]Shenzhen Key Laboratory of Biomimetic Robotics and Intelligent Systems, Department of Mechanical and Energy Engineering, Southern University of Science and Technology, Shenzhen 518055, China. [3]Guangdong Provincial Key Laboratory of Human-Augmentation and Rehabilitation Robotics in Universities, Southern University of Science and Technology, Shenzhen 518055, China. ✉e-mail: liuj9@sustech.edu.cn

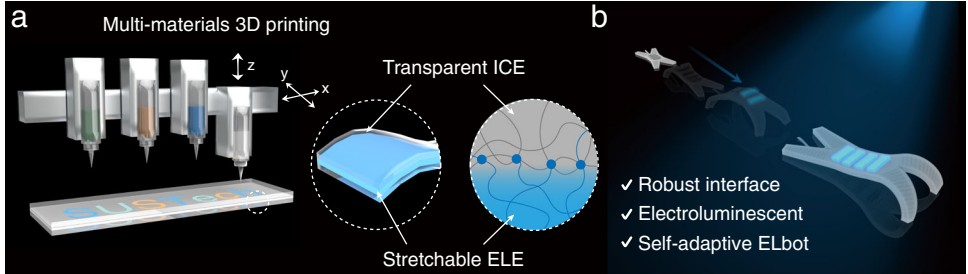

**Fig. 1 | 3D printing of electroluminescent devices and an integrated fabrication strategy for self-adaptive soft robots. a** Schematic illustration showing the multi-material direct ink writing of flexible EL devices. The electroluminescent devices consist of an electroluminescent elastomer (ELE) layer and an insulating dielectric elastomer (IDE) layer sandwiched between two ion conducting elastomer (ICE) layers. The robust interface is built through the covalent bonds. **b** Schematic illustration of the electroluminescent soft robot (ELbot) featuring chameleon-like adaptiveness to external light variation.

without compromising their unique electrical functionalities (i.e., ion conductivity, dielectric insulation, and electroluminescence). With the superior printability of our inks, high-fidelity printed 2D and 3D architectures, including a flexible wristband with a customized EL motif, were created. In addition, enabled by the ink formulation, the 3D printed EL devices display a high mechanical compliance and robust adhesion between the constituent layers, therefore permitting stable EL performance even under mechanical deformation. Our proposed strategy is readily integratable with other technological advancements, such as soft robotics. Through integrating our 3D printed EL devices with a pneumatic soft robot (Fig. 1b), we further demonstrate the creation of a chameleon-inspired self-adaptive soft robot, which can instantly change its surface color to match the environment. The facile and programmable fabrication strategy proposed here opens new avenues for creating the next generation flexible displays, wearable electronics, smart camouflages, and beyond.

## Results

### Ink design for direct ink writing

This study utilized direct ink writing (DIW), a widely-used extrusion-based 3D printing technique that is capable of constructing geometrically complex architectures through layer-by-layer deposition[21,22,26]. To fabricate a fully 3D printed electroluminescent device, which is composed of ion conducting elastomer (ICE), electroluminescent elastomer (ELE), and insulting dielectric elastomer (IDE) layers, we formulated various UV curable composite inks (Fig. 2a). Specifically, the ICE ink was composed of poly(acrylic acid) (PAA) supplemented with ionic monomers (i.e., 3-dimethyl(methacryloyloxyethyl) ammonium propane sulfonate, DMAPS) and an ionic liquid (i.e., 1-ethyl-3-methylimidazolium ethyl sulfate, EMES). The ELE ink was formulated by dispersing ZnS phosphor microparticles (50 μm in diameter) in a stretchable dielectric matrix material (i.e., thermoplastic PVDF-HFP elastomer, Dakin), of which the high matrix permittivity enables efficient use of the applied voltages via focusing the electric fields onto the phosphor microparticles[6,10]. Similarly, the IDE ink was formulated with the same composition as the ELE ink, but without the addition of the ZnS phosphor microparticles. PEGDA was also supplemented into the ICE, ELE, and IDE inks in order to make the 3DP structures polymerisable via UV irradiation, thus robust interface was built through covalent bonds (Fig. 2a). Finally, SiO₂ nanoparticles, a classic rheological modifier, were added to all three types of the inks (ICE, ELE and IDE inks), so that the inks can exhibit highly-desirable shear thinning and shear yielding properties[26,27]. These properties are particularly crucial to the success of the DIW 3D printing, which allow the inks to flow out from the fine nozzles smoothly, yet rapidly recovering its solid-like behavior to a sufficiently high storage modulus ($G'$) after extrusion for maintaining the prescribed shape[21].

As shown in Fig. 2c, d and Supplementary Figs. 1–2, the ICE, ELE, and IDE inks all acted like yield-stress materials and exhibited a pronounced shear thinning behavior with the viscosity decreased by four orders of magnitude as the shear rate increased from $10^{-2}$ to $10^{3}$ s⁻¹, facilitating the flow of material under extrusion force. The resultant rheological properties of our customized ELE inks enabled high resolution printing with good shape definition (Fig. 2e). To further promote the 3D printing resolution, we adopted an in situ UV cross-linking strategy, where the solidifying process can be initiated immediately and the inks can be continuously crosslinked after deposition (Supplementary Fig. 3), preventing sagging of the printed ink and hence enabling the fabrication of volumetric objects. Using this approach, inks with a lower $G'$ (i.e., ICE inks) exhibited a good print resolution (Supplementary Figs. 4–6), and can be readily transformed into intricate 2D and 3D structures upon UV irradiation. As demonstrated in Supplementary Fig. 7, a maze circuit, a spiral ring, a solid pyramid and a 3D suspended hollow pyramid made from the ICE inks were produced with good fidelity upon the optimization of printing parameters, such as the extrusion pressure, printing speed and chemical formulation, as shown in the printability evaluation diagrams (Supplementary Fig. 8). Apart from the distinct printability, our ICE, ELE and IDE inks offer long-lasting usability. They can be stored in dark at −4 °C over 1 month and room temperature for at least 1 week, without any significant change in their rheological properties and printability (Supplementary Figs. 9–10).

Mechanical compliance is a crucial requirement of flexible and stretchable EL devices[10,11]. In this respect, the 3D printed ICE, ELE, and IDE layers of our printable material system all exhibited high mechanical compliance (Fig. 2b and Supplementary Movie 1). Specifically, the ICE, ELE, and IDE layers exhibited a stretch of 720%, 500%, and 640%, and a strength of 1060 kPa, 344 kPa, and 290 kPa, respectively, under uniaxial stretching (Fig. 2f and Supplementary Figs. 11–13). It is also intriguing to note that the mechanical properties of the 3DP ICE samples were comparable to the properties of the samples produced using bulk molding, including Young's modulus ($E$), mechanical strength ($S$) and elongation at break ($\lambda$) (Fig. 2f).

Ionically conductive species are essential in EL devices[4–6]. Here, the 3DP ICE samples exhibited good ionic conductivity, attributed to the formation of specific ion-rich nano-channels that facilitated ion diffusion (i.e., EMES ionic liquid) in the complex of PAA and poly(DMAPS) polyzwitterions[28]. We observed a much higher ionic conductivity ($\sigma$) of the 3DP ICE samples compared to that of the bulk samples (Fig. 2f), which could be explained by the low cross-linking density within the 3DP samples that facilitated ion conducting. Notably, the electrical properties of our ICE materials appeared to be insensitive to stretching and cyclic deformation, suggesting the good robustness of our ICE inks. As shown in Supplementary Figs. 14–15, the ionic conductivity of the 3DP ICE samples remained almost constant when stretching the samples to a strain of 470%, and the resistance of the samples was approximately the same

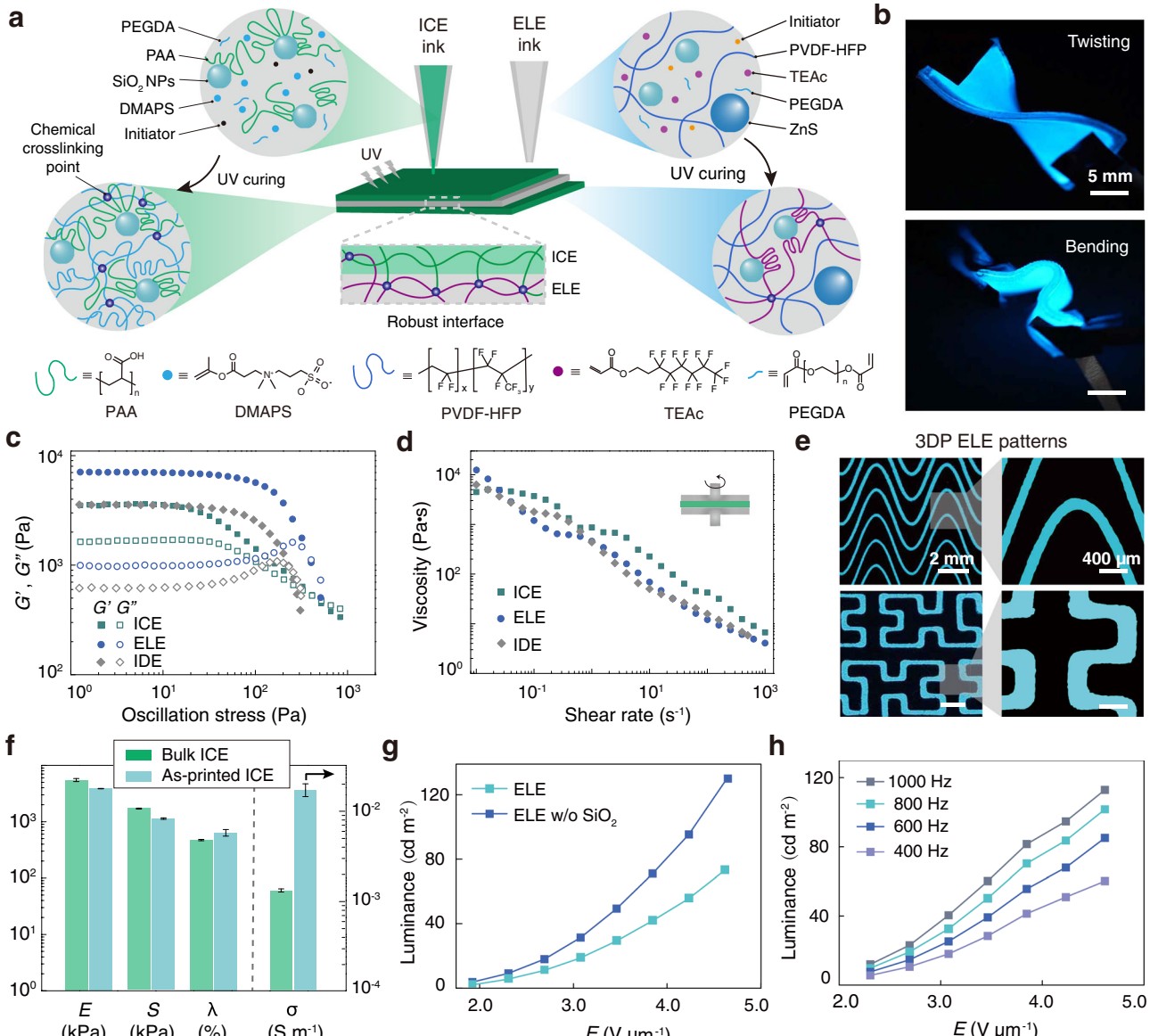

**Fig. 2 | Ink Design for Direct Ink Writing (DIW). a** Schematic illustration of the multi-material printing process for fabricating an EL device, which consisted of two ion conducting elastomer (ICE) layers and one electroluminescent elastomer (ELE) layer. The addition of SiO$_2$ nanoparticles in the inks leads to physically-crosslinked gels, which are fluidized by the shearing force during ink extrusion and can be recovered to the gel state right after printing. Further UV-initiated polymerization of DMAPS/PEGDA or TEAc/PEGDA leads to the formation of an intact network structure. **b** Images of the 3DP EL devices under mechanical deformation (i.e., twisting and bending), which are powered with an alternating current. Scale bar: 5 mm. **c** Shear storage moduli ($G'$) and loss moduli ($G''$) of the ICE, ELE, and IDE inks as a function of shear stress at 25 °C, measured under an oscillatory mode at a frequency of 1 Hz. **d** Apparent viscosity of the optimized ICE, ELE, and IDE inks as a function of shear rate at 25 °C. **e** Images of the high-fidelity electroluminescent prints of the ELE patterns. **f** Summary of the physical parameters (i.e., modulus ($E$), strength ($S$), elongation at break ($\lambda$) and conductivity ($\sigma$)) of the ICE samples fabricated from both molding and the 3D printing techniques. **g** Plotting of the luminance intensity of the 3DP ELE patterns using inks with or without SiO$_2$ NPs against the applied voltage per thickness ($E$) (thickness of ELE layer was 150 μm, frequency of the applied voltage was 500 Hz). **h** Plotting of the luminance intensity of the 3DP ELE patterns against the applied voltage per thickness ($E$) at various frequencies of the applied voltage. Data in **f**, **g**, and **h** are means ± S.D., $n = 3$ independent samples.

over multiple cycles at a strain of 100%. It is noted that humidity might have an effect on the electrical stability of the ICE samples (Supplementary Fig. 16). To mitigate this problem, the sample can be coated with an elastic polymer, such as PDMS, to achieve a stable electrical performance in high humidity environment (Supplementary Fig. 17). Lastly, our ICE samples featured a good optical transparency that is highly desirable for EL devices (Supplementary Fig. 18)[29].

Next, we investigated the influence of the dielectric parameters of the ELE layers on the light emission performance of the EL devices. The dielectric constants of the PVDF-HFP elastomer (IDE) and the PVDF-

HFP/ZnS (IDE/ZnS) nanocomposites were 8 and 8.6, respectively, at 1 kHz (Supplementary Fig. 19). This exceptional polarizability is ascribed to the base matrix (PVDF-HFP elastomers) of the ELE ink, while addition of SiO$_2$ and PEGDA did not pose significant effect on the overall polarizability. By gradually increasing the amplitude of the applied square wave voltage, the luminance and the emission intensity were rapidly increased (Fig. 2g and Supplementary Figs. 20–21). This voltage-dependent emission characteristic presents a trade-off between the brightness and the applied voltage intensity. Although the presence of SiO$_2$ nanoparticles in the ELE layer slightly reduced the luminance (20 cd m$^{-2}$ in ELE inks with SiO$_2$ vs 30 cd m$^{-2}$ in ELE inks

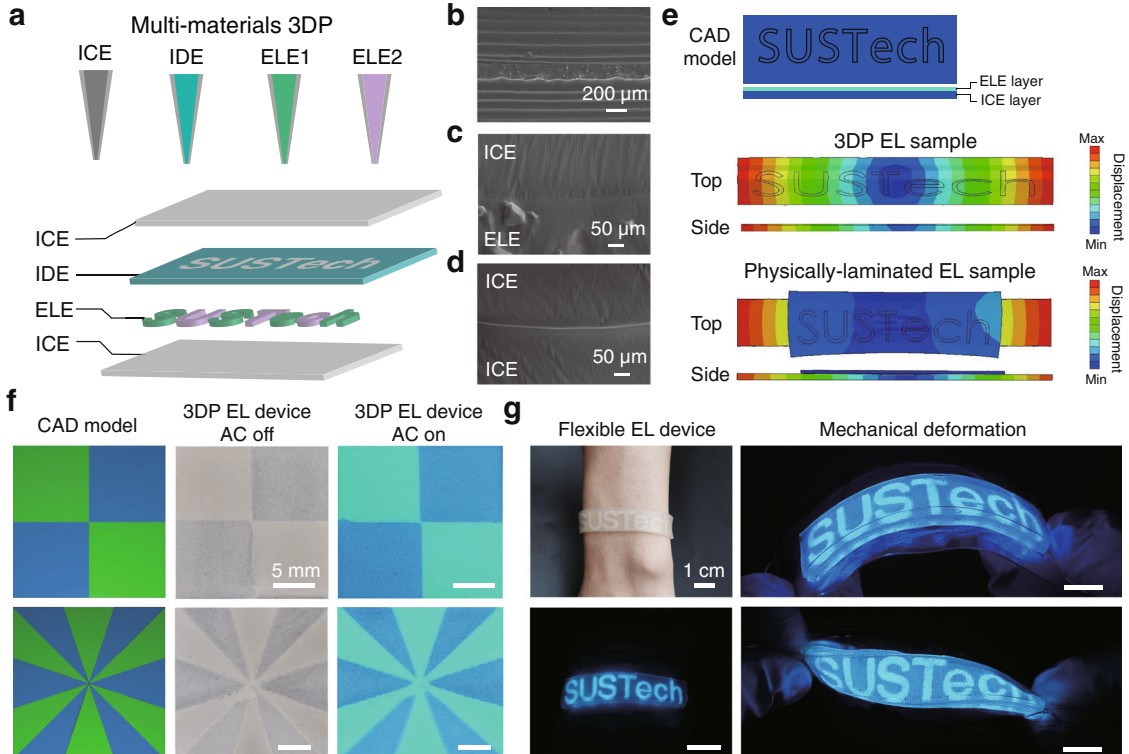

**Fig. 3 | Multi-functional 3D printing. a** Schematic illustration of the multi-material DIW process for fabricating EL devices. ICE, IDE and ELE Inks were printed sequentially. **b**–**d** SEM images of the multi-layer structure of the 3DP samples (**b**), the interface between the ICE and ELE layers (**c**), and the interface between two ICE layers (**d**) of the 3DP samples. Scale bar: 200 μm (**b**) and 50 μm (**c**, **d**). **e** Simulated displacement distributions within the 3DP EL device (top) and a physically-laminated sample (bottom) at an overall applied strain of 35%. Both devices consisted of a ELE layer (with "SUSTech" logo) and an ICE layer. The simulation was performed *via* finite element analysis. Mechanically compliant and robust interface was evidenced in the 3DP EL device, while weak interface and interfacial delamination were detected for the physically-laminated EL device. **f** The designated CAD models and the images of the 3DP EL devices with multi-color electroluminescent components with alternative voltage on or off. Scale bar: 5 mm. **g** Images of a 3DP flexible EL wristband, indicating the stable electroluminescent features under mechanical deformation, such as bending and twisting. Scale bar: 1 cm.

without SiO$_2$ at 3 V m$^{-1}$), a luminance of 20 cd m$^{-2}$ of the 3DP ELE layer is considerably high and could be easily identified in dim lighting environments. Moreover, the luminance could be further increased up to 120 cd m$^{-2}$ (Fig. 2g, h and Supplementary Fig. 22) by rationally increasing the AC voltage (i.e., 4.5 V m$^{-1}$) and/or AC frequency (i.e., 1000 Hz), satisfying the brightness requirement for regular indoor lighting.

**Multi-material 3D printing of electroluminescent devices**

One of the appealing characteristics of DIW is its multi-material printing capability, hence enabling the integration of various voxelated components into one single construct and/or device[21,22,26]. We demonstrated this capability via fabricating a model architecture that encompassed distinct components of the ICE ink, IDE ink, and ELE inks with ZnS phosphors in different colors (Fig. 3a). In brief, an ICE layer was first printed as the conductive substrate, followed by printing the ELE patterns using the ELE inks. To insulate the ELE inks, an IDE ink was then deposited onto the surrounding cavities that were uncovered by the ELE ink. The device was then finished with printing another ICE layer on top (Supplementary Fig. 23, see detailed information in the Experimental Part).

Robust interfacial adhesion between distinct 3D printed features is a crucial prerequisite for its ability to effectively withstand structural deformation without delamination, hence enabling a stable EL performance[21]. Attributed to the interfacial chain entanglement and formation of covalent bonds within the multi-layered EL samples upon UV curing during the 3DP process (Fig. 2a), robust interfaces between layers of the samples were obtained. From the SEM and confocal images (Fig. 3b–d and Supplementary Fig. 24), a well-defined clear and coherent adhesion was observed at both the homogeneous ICE/ICE interface and the heterogeneous ICE/ELE interface of the 3D printed multi-layered constructs. Quantified by 180-degree peeling tests, the interfacial toughness of the ICE-ICE, ICE-ELE, and ICE-IDE interfaces reached as high as 670 J m$^{-2}$, 150 J m$^{-2}$, and 30 J m$^{-2}$, respectively (Supplementary Figs. 25–26). Likewise, our simulated results from finite element analysis show that the mechanical strain can be efficiently redistributed among the ICE, ELE, and IDE layers through the robust interfacial bonding, therefore interfacial delamination can be desirably avoided (Fig. 3e and Supplementary Fig. 27, see detailed information in the Experimental Part).

We then designed and fabricated two EL devices with distinct blue and green electroluminescent patterns through multi-material 3D printing (Fig. 3f). The color of the 3DP patterns was greyish, yet vibrant luminescence was observed upon the activation by an AC electric field. We further demonstrated the capability of our material choice in creating a flexible and wearable EL device via fabricating a flexible wristband with an electroluminescent "SUSTech" logo (Fig. 3g and Supplementary Movie 2). The as-printed EL wristband exhibited compliant mechanical properties, thus the luminescent patterns were stable even under different modes of mechanical deformation, such as bending, twisting, and stretching. It should also be noted that the shape of the device could be completely recovered after the deforming force was removed (Supplementary Movie 1). Overall, we suggest a generable yet facile strategy that can be used to effectively transform conventional electroluminescent components into soft, stretchable, customizable, and stable EL devices.

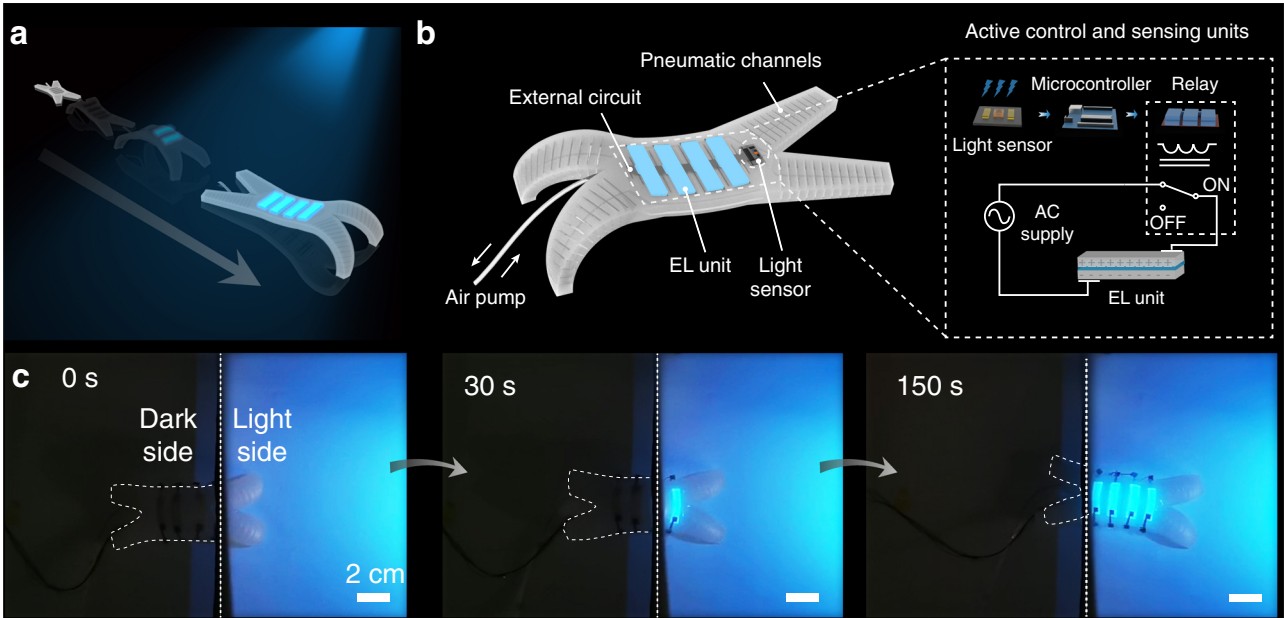

**Fig. 4 | Control logic and the spatially color-changing ability of the ELbot.**
**a** Schematic illustrating the color-matching strategy of the ELbot, in imitation of the color-changing ability of chameleons. **b** Control logic of the ELbot. **c** Instantaneous color-changing ability of the ELbot. The corresponding EL devices on the soft robot instantly illuminated when they were exposed to a blue light. Scale bar: 2 cm.

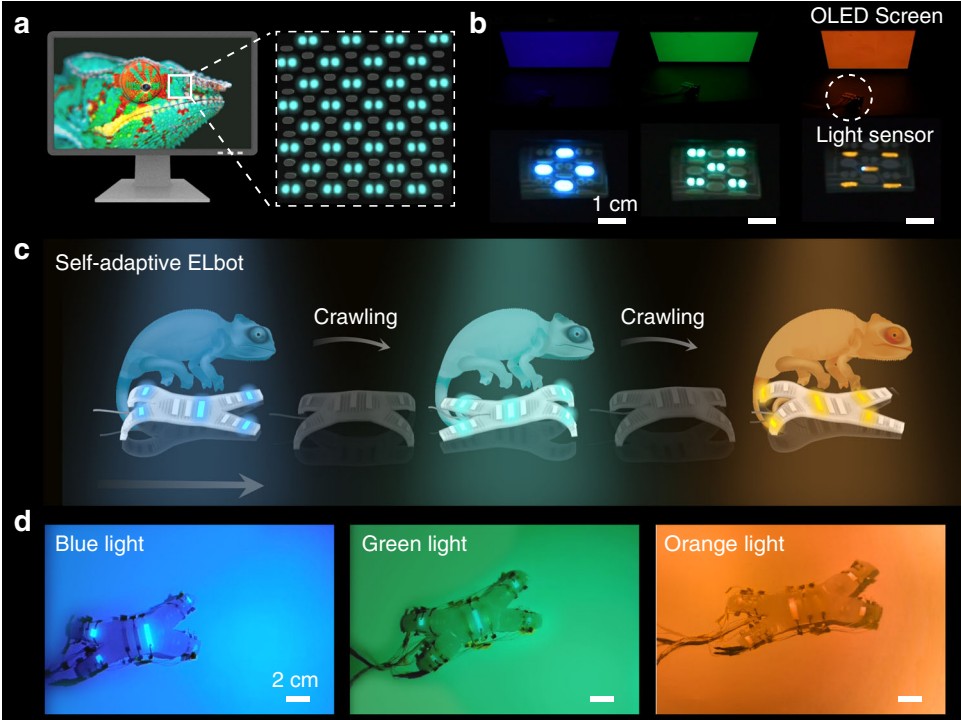

**Fig. 5 | Self-adaptive background color-matching ability of the ELbot.**
**a** Schematic illustration of a self-adaptive color-matching EL display. The chameleon picture is adapted from Pixabay under CC0 Creative Commons License. **b** Instantaneous color-changing of the EL-integrated display in response to the background light variation. Blue, green, and orange light-emitting EL units were printed on the display. A 200-V AC power voltage at a frequency of 1000 Hz was set. Scale bar: 1 cm. **c** Schematic illustrating the chameleon-inspired background-matching strategy of the ELbot. The chameleon sketches are adapted from Freepick under the CC0 Creative Commons License. **d** Instantaneous color-changing of the ELbot in response to the background light variation. Scale bar: 2 cm.

## Artificial camouflages through integrating 3DP EL devices with a soft robot

Nature is replete with living species that can self-adapt the skin color in response to the environment variation, such as chameleons. Their self-adaptive color-changing behavior involves the collection of environmental information through retinas, and corresponding control of the chromatophores in their skin via neural systems[29–31]. Inspired by these creatures, we created artificial camouflages by integrating the 3DP EL devices with a pneumatic soft robot (termed as ELbot hereafter, Fig. 4a). Our ELbot is expected to retrieve the

background color and change its spatial surface color to match the environment in an autonomous fashion, in imitation of the concealing coloration of natural camouflages.

Supplementary Figs. 28–29 show the design schematics and the fabrication procedure of the ELbot. We incorporated three main constituents within the ELbot: a quadrupedal robotic 'walker' created by molding as the motion unit, EL units printed on the quadrupedal walker, and a control unit for sensing and implementing actuation. To enable fast crawling locomotion, the quadrupedal walker contained mesofluidic pneumatic channels that were repeatedly inflated and deflated (Supplementary Fig. 30 and Supplementary Movie 3)[9,27,32]. Fig. 4b and Supplementary Fig. 31 show the control logic of the background matching strategy implemented in the ELbot. To achieve autonomous background matching, the ELbots were integrated with a light sensor that can retrieve the information of the environmental color, and output the estimated RGB value. The AC supplies to the EL units were then individually switched on/off according to the interpreted environmental RGB intensity or the background light wavelength using a microcontroller that controlled the circuit via relays.

We first fabricated an ELbot that integrated four on-broad EL devices printed using the blue light-emitting ELE ink to demonstrate the background matching ability of the ELbot on a spatial scale (Fig. 4c). To achieve a strong adhesion between the soft robot and the 3DP EL devices, the surface of the soft robot was treated with TMSPMA prior to printing. Owing to the superior flexibility of the EL devices, the 3DP EL devices can accommodate and conform to the dynamic surface of the soft robot, which was persistently deformed by the repeated inflation-deflation cycles. As the robot crawled to a hostile environment (the blue light habitat here), the EL devices emitted a blue light and instantly blended in the background environment without any notable delay (Supplementary Movie 4).

In similar fashion, multiple color matching within one single device could also be achieved via printing different EL devices through the multi-material DIW printing. To demonstrate this concept, we first fabricated a self-adaptive color-matching display via 3D printing green, blue and orange light-emitting EL units on a PET sheet (Fig. 5a, b and Supplementary Movie 5). Upon exposure to distinct light environments, the corresponding 3DP EL units on the display instantly illuminated and emitted light with a similar color. Our self-adaptive color-matching strategy can work in concert with soft robotic systems to enable a biomimetic motion camouflage (Fig. 5c). We fabricated an ELbot that embodied a combination of 3D printed green, blue and orange light-emitting EL units on-broad. As illustrated in Fig. 5d and Supplementary Movie 6, when the robot crawled to three distinct habitats of blue, green and orange light, the EL devices on the ELbot were selectively activated and emitted a background-matching light via the light-sensing and circuit control system. This capability enables the potential to recapitulate the innate camouflaging behavior in some living species.

## Discussion

In summary, we present a streamlined approach for fabricating mechanically compliant and flexible EL devices through developing high-performance 3D printable EL inks. Enabled by the good interfacial robustness of the 3DP EL devices, the 3DP EL devices exhibited stable electroluminescent performance even under mechanical deformation. Through integrating the 3D printed EL devices with a soft robot, a chameleon-inspired artificial camouflage was designed and fabricated to demonstrate the potential of our 3DP EL devices in automated background-matching applications.

Differing from previous artificial camouflages made of shape-memory polymer[33] or stimuli-responsive polymer[34], of which the response is slow and the reversibility is poor upon external stimuli[29], our ELbot is able to attain perceivably instant response and reversible color transformation with comparable timescale to physiological

response through actively controlling the AC voltage supply in harmony with the external background color. This instant matching capability is ideal for camouflage applications[9,29]. It should be pointed out that the above demonstration shows the ability of our ELbot in response to a specific monochromatic background. Although not demonstrated in this study, to enable response to unlimited background color, the EL unit can be printed in a pixel-wise arrangement, where each pixel is made of three subpixels (red, green, blue ELE inks) to yield a complete coverage of the RGB color gamut. In addition, spatially color-matching response could also be readily realized through programming each pixel according to the local environment information.

Looking ahead, the multi-material 3D printing strategy proposed here offer advantages of rapid prototyping and customizability in the EL device fabrication. With the proposed 3D printing strategy, intricate EL devices can be custom-made for specific applications, opening up new avenues for the next generation of completely soft light-emitting devices, smart displays, and camouflaging systems.

## Methods

### Ink composition and preparation

To fabricate a fully 3D printed electroluminescent device, three different UV-curable composite inks were formulated, specifically for the ion conducting elastomer (ICE) layer, electroluminescent elastomer (ELE) layer and insulting dielectric elastomer (IDE) layer. The ICE ink was composed of poly(acrylic acid) (PAA) supplemented with ionic monomers (i.e., 3-dimethyl(methacryloyloxyethyl) ammonium propane sulfonate, DMAPS) and an ionic liquid (i.e., 1-ethyl-3-methylimidazolium ethyl sulfate, EMES). There were three ICE ink recipes used in our work, namely ICE1, ICE2, and ICE3. For ICE1, the weight percentages of PAA, DMAPS, and EMES in the ink were set at 12.4 wt.%, 61.7 wt.%, and 25.9 wt.%. For ICE2, the corresponding weight percentages were 6.6 wt.%, 65.8 wt.% and 27.6 wt.%. For ICE3, the corresponding weight percentages were 4.5 wt.%, 67.2 wt.%, and 28.3 wt.%. These inks were created for comparing their mechanical properties as shown in Supplementary Fig. 9, and ICE1 was typically used in the study. Next, the crosslinker PEGDA400 (0.8 wt.%), photo-initiator I2959 (0.5 wt.%) and silica (5, 10, or 15 wt.%) were added into the PAA/DMAPS/EMES mixture solution in methanol under 3-min vigorous mixing in a planetary mixer (AR-100, Thinky) at 2000 rpm. The solvent was removed through evaporation, followed by defoaming for 2 min at 2200 rpm using a planetary mixer. The as-obtained ICE ink was stored in dark prior to use.

The IDE ink was prepared by dissolving 0.2 g PVDF-HFP in 8 mL TEAc under magnetic stirring for 12 h. The crosslinker PEGDA400 (1 wt.%), photo-initiator TPO (1 wt.%) and silica nanoparticles (5 wt.%) were added into the viscous paste, and mixed vigorously within the planetary mixer at 2500 rpm for 1 min, giving the insulating dielectric ink. The ELE ink was prepared by adding ZnS phosphors (40 wt.%) into the insulating dielectric ink, under 3-min vigorous mixing in the planetary mixer at 2000 rpm, following by defoaming at 2200 rpm for 1 min. Blue, green and orange light-emitting ELE inks were prepared with the use of ZnS particles doped with different transition metals. All these freshly-prepared inks for DIW were loaded to a UV-blocking 5-mL syringe barrel (EFD Nordson), and stored in dark prior to further 3D printing.

### 3D printing procedure

3D printing of the EL devices was conducted on a 3D Bio-Architect multi-nozzle workstation (Regenovo). To achieve stable 3D printing with continuous and uniform filaments, we first performed a printability study to optimize the printing parameters (i.e., printing speed and printing pressure) for the ICE, IDE, and ELE inks. The optimized parameters are summarized below. The ICE ink was extruded through a nozzle (0.26 mm in diameter) at a printing speed of 12 mm s$^{-1}$ under a

pressure of 0.3 MPa at room temperature; The IDE ink was extruded through a nozzle (0.21 mm in diameter) at a printing speed of 10 mm s⁻¹ under a pressure of 0.5 MPa; The ELE ink was extruded through a nozzle (0.21 mm in diameter) at a printing speed of 5 mm s⁻¹ under a pressure of 0.5 MPa.

To fabricate flexible EL devices, the 3D printed structures were first designed using Solidworks (Dassault Systemes), and were then converted into G-code. The ICE, IDE, and ELE inks within three individual UV-blocking syringe barrels (5 mL) were loaded to the workstation. An ICE layer was first printed as the substrate, followed by printing ELE patterns using one or more ELE inks with the surrounding edges filled with the IDE ink (Supplementary Fig. 23). The device was then finished up with printing another ICE layer on top. During 3D printing, UV radiation (365 nm wavelength, 400 mW cm⁻² and power intensity of 50%) was applied in situ to simultaneously cure the 3D printed structures. After printing, the as-printed devices were further cured for 60 s under UV radiation.

### Fabrication of quadrupedal soft robots

The pneumatically-actuated quadrupedal robots were fabricated following previously described protocols deploying the soft lithography strategy[9,32]. They were prepared by casting PDMS precursor into a customized mold designed with SolidWorks (Dassault Systemes). The mold consisted of an upper mold and a lower mold, and were printed with a FDM 3D printer (Ultimaker S5). Specifically, the top layer with air channels was prepared by casting a flexible silicone rubber (Ecoflex 00-30, Smooth-on, Inc) into the upper mold (Supplementary Fig. 28), while the bottom layer was prepared by casting a rigid PDMS precursor (Sylgard 184, Dow Corning) into the lower mold. After degassing in a vacuum drier (Sciencetool, DV-9252), the Ecoflex precursor was partially cured under room temperature for 1 h, and the PDMS precursor was partially cured at 75 °C for 20 min. To achieve a robust interface between the top and bottom layers, the partially cured top layer was attached to the partially-cured bottom layer, and the assembly was further cured at 75 °C for another 1 h. Five soft and flexible silicone rubber tubes (1 mm in outer diameter) were then inserted into the designed air channels in the assembly and were further sealed with Sylgard 184 PDMS precursor under 75 °C for 1 h. To achieve independent control of the four feet and the central body of the quadrupedal soft robots, air was pumped to each tube separately via syringe manipulation. Movements, such as wriggling and crawling, were driven under alternating inflation and deflation of the quadrupedal soft robots (Supplementary Fig. 30).

### Fabrication and assembly of the ELbots

To enhance the interfacial covalent bonding between the silicone elastomer of the quadrupedal soft robot and the EL units printed in the following step, the surface of the quadrupedal soft robot was first functionalized with 3-(trimethoxysilyl)propyl methacrylate (TMSPMA) silane. Specifically, the as-prepared robot was thoroughly cleaned with de-ionized water and ethanol in sequence, followed by complete drying under nitrogen flow. Surface of the robot was activated with oxygen plasma (30 W, 200 mtorr; Harrick Plasma PDC-002, 1 min), prior to 2 h incubation in the TMSPMA silane solution (4.15 g TMSPMA in 100 mL ethanol). After surface treatment, the soft robot was fixed on the DIW platform. The ICE, ELE, and IDE inks were then printed on the soft robot via multi-material 3D printing using the aforementioned method, as shown in Supplementary Fig. 29. Once the 3D printing process was completed, the EL soft robot was exposed to UV radiation for another 600 s for complete curing. The top and bottom ICE layers were then connected to copper wires (0.8 mm in diameter) using a conductive Ag paste. Two quadrupedal soft robots were fabricated. The first quadrupedal soft robot was embodied with four EL units that can emit blue light, and the second robot was embodied with various EL units that can emit either blue, green or orange light.

A commercially available light sensor (AS7341, ams AG, 20 × 20 mm) was used as the light sensing system, which was placed to the soft robots at the position as indicated in Fig. 4b.

### Control logic of the ELbots

The control logic of the EL integrated soft robots is shown in Fig. 4b. To implement spatially background matching, the control of the first ELbot (with four blue light-emitting EL units) relied on the detection of the light intensity of the background using the light sensor. A microcontroller (Arduino UNO) was used to control the on/off of the relay of each EI unit according to the light intensity, which in turn switching on/off the AC supply (150–600 V) of the corresponding EL unit. On the other hand, the control of the second ELbot (with various light-emitting EL units) relied on the detection of the light wavelength, where the corresponding relay of the EL unit was independently switched on/off in accordance with the measured wavelength of the background environment.

### Fabrication and control logic of the real-time adaptive multi-color EL display

Blue, green, and orange-light emitting EL units were printed on a 50-μm PET sheet in an arranged configuration, as illustrated in Supplementary Fig. 31, using the aforementioned multi-material printing method. Similar to the control logic used in the ELbots, a light sensor (AS7341, ams AG, 20 × 20 mm) was placed near the display for measuring the wavelength of the background light, and an Arduino microcontroller was used to control the on/off of the EL units that can emit specific colors according to the background light wavelength.

## Data availability

The data that support the findings of this study are provided in the source data file. Source data are provided with this paper.

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

## Acknowledgements

J.L. acknowledges the financial support from Natural Science Foundation of Guangdong Province (2022A1515010152 and 2020A1515110288), Basic Research Program of Shenzhen (JCYJ20210324105211032 and RCBS20210609103713046), and MechERE Centers at MIT and SUSTech (Y01346002). This work was also supported in part by the Science, Technology and Innovation Commission of Shenzhen Municipality (ZDSYS20200811143601004) and the Additive Manufacturing Innovation Center at SUSTech. The authors also would like to acknowledge the technical support from SUSTech Core Research Facilities, Prof. Hong Wang and Prof. Kai Wang for their generous access to dielectric and luminance measurement facilities, respectively.

## Author contributions

J.L. conceived the ideas. J.L. and P.Z. designed the experiments. P.Z., I.M.L., G.C., J.S.L., X.C., J.Z., C.C., X.L., and J.L. performed experiments and analysed the experimental data. J.L. and P.Z. wrote the manuscript with input from all authors. J.L. supervised the study.

## Competing interests

The authors declare no competing interests.
