## [Peer Review File · Nature Communications]

Integrated 3D Printing of Flexible Electroluminescent Devices and Soft RobotsREVIEWER COMMENTS

Reviewer #1 (Remarks to the Author):

The manuscript presents the the fabrication of an almost fully functional soft robot in one shot 3D printing process. It consists of a highly conducting elastomer as the electrodes, a dielectric elastomer as the insulating layer, and a ZnS phosphors-loaded elastomer as the electroluminescent layer. All the prerequisite properties for printable ink were tailored and tested, EL properties and intensity were also examined, and finally, a fully operating 3D printed soft robot was presented. To further describe the advantages of soft robotics, ELbot was printed, followed by attaching a light sensor, and controlled luminesce, showing a color-matching ability to the surroundings. The manuscript is well written and presents a nice approach. It can be considered for publication in Nature Communication after addressing the following minor comments:

- a. Is there any change in the printed structure after being exposed to air? The PEGDA and PAA may adsorbs water/humidity.
- b. What is the print life ("open time") of the inks at room temperature?
- c. Typo: the word "showing" is repeated twice in Fig 1 caption.

Reviewer #2 (Remarks to the Author):

This article demonstrates an interesting topic of Flexible EL devices integrated with Soft Robots. The authors not only provides comprehensive characterization, but also the extensive performance evaluation with impressive supplementary videos. This proposed research and investigation will be of great interests to materials chemist and soft robot scientists. Therefore, I would recommend it to be accepted after the following technical issues to be addressed.

1. More detail information for the ICE Ink preparation, such as ? wt.% of PAA and DMAPS rater than just molar ratio.
2. What does this mean - "... with the surrounding edges filled with the IDE ink." in page 14, line 2? Any Figure illustration to demo this step?
3. Why choose quadrupedal robots as the demo device? Any specific reason?
4. For EL devices, the thickness of the ICE layers, which was used to sandwich the EL sample layer, will affect the performance or not?
5. Any tests on visible light (> 450 nm) besides single-colour light?

Point-by-point responses to the reviewers' comments (NCOMMS-22-11658)

Reviewer #1

General comment. The manuscript presents the fabrication of an almost fully functional soft robot in one shot 3D printing process. It consists of a highly conducting elastomer as the electrodes, a dielectric elastomer as the insulating layer, and a ZnS phosphors-loaded elastomer as the electroluminescent layer. All the prerequisite properties for printable ink were tailored and tested, EL properties and intensity were also examined, and finally, a fully operating 3D printed soft robot was presented. To further describe the advantages of soft robotics, ELbot was printed, followed by attaching a light sensor, and controlled luminescence, showing a color-matching ability to the surroundings. The manuscript is well written and presents a nice approach. It can be considered for publication in Nature Communication after addressing the following minor comments:

Answer: We appreciate the reviewer's time for his/her constructive comments on our manuscript. We have now addressed the comments, and a detailed point-by-point response to all comments can be found below.

Comment 1. Is there any change in the printed structure after being exposed to air? The PEGDA and PAA may adsorb water/humidity.

Answer: Following the reviewer's comment, we have now further provided information regarding the change in the printed ICE structure after being exposed to high-humidity condition (**Figure R1a**). The ICE ink is composed of PAA, DMAPS, EMES, silica, PEGDA400 and photo-initiator I2959. As shown in **Figure R1b**, the dimension of the 3D printed filaments remains almost unchanged, with a less than 5% increase in dimension after being exposed to high-humidity condition for six days, which could be caused by swelling or sagging of the filament due to gravity. This finding suggests the sufficient stability of the printed ICE structure and the insignificant swelling effect of PEGDA and PAA using our ICE ink formulation.

Figure R1 | (a) Schematic illustration of the experimental setup for exposing the 3DP ICE samples to high-humidity condition. (b) The change in the size of the 3D printed ICE filament at different days.

Moreover, we monitored the evolution of resistance of the ICE samples (rectangular shape with 20 mm in length, 8 mm in width, and 1 mm in thickness) upon exposure to high humidity environment (a sealed box was filled with supersaturated potassium chloride solution) for 6 days. As shown in **Figure R2**, the resistance of original ICE sample decreased by 23% at Day 1 due to the absorption of moistures. Afterwards, the resistivity remains stable in the following 5 days. To reduce the moisture sensitivity of the samples, we could simply sealed the ICE sample with another moisture-nonsensitive elastomer, such as PDMS. As shown, the resistance experienced a slight variation in the following 6-day exposure to high humidity environment.

Figure R2 | Evolution of conductivity of the ICE samples and the ICE samples encapsulated in PDMS when stored

stored at high humidity environment.

Comment 2. What is the print life (“open time”) of the inks at room temperature?

Answer: We thank the reviewer for this comment. In our previous manuscript, we have evaluated the shelf life of the inks stored in dark at $-4\text{ }^{\circ}\text{C}$ over one month (**Supplementary Fig. S9**), which shows no significant change in their rheological properties and printability. Following the reviewer’s comment here, the rheological properties and printability of the inks stored at room temperature were now evaluated. **Figure R3** shows the rheological profiles of the three inks (ICE, ELE, and IDE inks) upon storage in dark at room temperature ($25\text{ }^{\circ}\text{C}$) over one week. Nearly no change in the rheological properties was observed, indicating that the printability of the inks remains stable over long-term storage.

Figure R3 | The rheological profiles of the inks before and after one-week storage at room temperature.

Comment 3. Typo: the word “showing” is repeated twice in Fig 1 caption.

Answer: We thank the reviewer for pointing this out, and it has now been corrected.

Reviewer #2

General comment. This article demonstrates an interesting topic of Flexible EL devices integrated with Soft Robots. The authors not only provide comprehensive characterization, but also the extensive performance evaluation with impressive supplementary videos. This proposed research and investigation will be of great interests to materials chemist and soft robot scientists. Therefore, I would recommend it to be accepted after the following technical issues to be addressed.

Answer: We thank the reviewer for appreciating the originality of our work and recommending the publication of our work in *Nature Communications*.

Comment 1. More detail information for the ICE Ink preparation, such as ? wt.% of PAA and DMAPS rather than just molar ratio.

Answer: Following the reviewer's suggestion, we have now revised the 'ink composition and preparation' section in Methods. "The molar ratio of PAA to DMAPS..." was now replaced by "*There were three ICE ink recipes used in our work, namely ICE1, ICE2, and ICE3. For ICE1, the weight percentages of PAA, DMAPS, and EMES in the ink were set at 12.4 wt.%, 61.7 wt.% and 25.9 wt.%. For ICE2, the corresponding weight percentages were 6.6 wt.%, 65.8 wt.% and 27.6 wt.%. For ICE3, the corresponding weight percentages were 4.5 wt.%, 67.2 wt.% and 28.3 wt.%.*", as highlighted in yellow in our revised manuscript.

Comment 2. What does this mean - "... with the surrounding edges filled with the IDE ink." in page 14, line 2? Any Figure illustration to demo this step?

Answer: In our fabrication procedure of EL devices, after printing an ICE layer as the device substrate, the designated ELE patterns were printed using the ELE inks. In order to insulate the ELE pattern, the IDE ink was then deposited onto the substrate to fill the cavities that is uncovered by the ELE ink. To better clarify our fabrication procedure, we have now amended the sentence to '*In brief, an ICE layer was first printed as the conductive substrate, followed by printing the ELE patterns using the ELE inks. To insulate the ELE inks, an IDE ink was then deposited onto the surrounding cavities that were uncovered by the ELE ink.*', as highlighted in yellow. We have also drawn a new scheme (**Figure R4**) to demonstrate the full process of 3D

printing of a “SUTech” logo through multi-material printing.

Figure R4 | Schematic illustrating the 3D printing of a “SUTech” logo through multi-material printing.

Comment 3. Why choose quadrupedal robots as the demo device? Any specific reason?

Answer: Quadrupedal robots are a classic design of locomotive soft robots. Owing to its simplicity and capability of performing complex motions, this class of design has been widely adopted for demonstrating the technological advancements in the field of soft robotics (*PNAS*, **2011**, 108, 20400-20403; *Science*, **2012**, 337, 828-832). Several benefits can result from the use of quadrupedal robots as the demo device here. The primary benefit is that this opens up the creation of flexible electroluminescent robots targeted in this study as the design of quadrupedal robots enables the fabrication of entirely soft robots without the need of any rigid skeleton (*PNAS*, **2011**, 108, 20400-20403). In addition, the use of a widely adopted design as the demo device will encourage the adoption of our technology in the research community, therefore further prompting the integration of our technology with other existing soft robotic technology that built upon the design of quadrupedal robots.

Comment 4. For EL devices, the thickness of the ICE layers, which was used to sandwich the EL sample layer, will affect the performance or not?

Answer: The thickness of the ICE layers indeed have an influence on the performance of the 3D printed EL devices. As shown in **Figure R5a**, the resistance of the ICE layers decreased with the increasing number of

the ICE layers (the thickness of each layer is $\sim 200 \mu\text{m}$). After about four layers, the resistance reached a plateau value, indicating that lower resistance will not be encouraged by further increasing the number of layers. Therefore, all EL devices demonstrated in this study were fabricated with five ICE layers. In addition to the resistance of the ICE layers, the thickness of the ICE layers also dominates the electroluminescence intensity of the EL devices and thus the applied voltage required. As illustrated in **Figure R5b**, the electroluminescence intensity increased with the increasing number of ICE layers. Hence, lower applied voltage is needed to obtain the same electroluminescence intensity of the EL device.

Figure R5 | **(a)** Resistance of the printed ICE samples with different number of layers. Each layer is $\sim 200 \mu\text{m}$. **(b)** The effect of the ICE layer thickness (controlled by the number of layers) and the applied voltage on the electroluminescence intensity of the EL devices.

Comment 5. Any tests on visible light ($> 450 \text{ nm}$) besides single-colour light?

Answer: All the EL devices fabricated in our study emit visible light ($> 450 \text{ nm}$). In our manuscript (**Figure 3f** and **Figure 5**), we have demonstrated that the devices are able to emit blue, green and orange light with the use of different types of ZnS phosphor microparticles in the EL inks. It is also possible to have red light-emitting EL devices, however, due to safety concern about the use of heavy metal Cd in red light-emitting ZnS phosphor, we did not demonstrate this in our manuscript but the fabrication procedure is literally the same. In addition, to yield a device that can emit any wavelength of light in the visible spectrum, the EL device can potentially be printed in a pixel-wise arrangement, where each pixel is made of the red, green and blue subpixels to cover the entire RGB gamut. We have discussed this possible work in our revised manuscript and exploring this will be of interest in future work, as highlighted in yellow.

REVIEWERS' COMMENTS

Reviewer #1 (Remarks to the Author):

The authors have addressed well the comment, and the paper is suitable for publication

Reviewer #2 (Remarks to the Author):

The authors have fully addressed reviewers' comments dot-by-dot. Therefore, I would recommend it to be considered for publication without further revision.

A point-by-point response to the reviewers' comments (Manuscript NCOMMS-22-11658A)

Reviewer #1

Comment: The authors have addressed well the comment, and the paper is suitable for publication.

Answer: We thank the reviewer for appreciating the improvement of our revised manuscript.

Reviewer #2

Comment: The authors have fully addressed reviewers' comments dot-by-dot. Therefore, I would recommend it to be considered for publication without further revision.

Answer: We thank the reviewer for appreciating the improvement of our revised manuscript.